# The Interplay between Neurotransmitters and Calcium Dynamics in Retinal Synapses during Development, Health, and Disease

**DOI:** 10.3390/ijms25042226

**Published:** 2024-02-13

**Authors:** Johane M. Boff, Abhishek P. Shrestha, Saivikram Madireddy, Nilmini Viswaprakash, Luca Della Santina, Thirumalini Vaithianathan

**Affiliations:** 1Department of Pharmacology, Addiction Science, and Toxicology, College of Medicine, University of Tennessee Health Science Center, Memphis, TN 38163, USA; johane.boff@uthsc.edu (J.M.B.); drabhishekshrestha@gmail.com (A.P.S.); 2College of Medicine, The University of Tennessee Health Science Center, Memphis, TN 38163, USA; smadired@uthsc.edu; 3Department of Medical Education, College of Medicine, University of Tennessee Health Science Center, Memphis, TN 38163, USA; nviswapr@uthsc.edu; 4College of Optometry, University of Houston, Houston, TX 77204, USA; 5Department of Ophthalmology, Hamilton Eye Institute, University of Tennessee Health Science Center, Memphis, TN 38163, USA

**Keywords:** retina, neurotransmitters, calcium, neural processes, vision, retinal development, synaptic transmission, retinal diseases, ocular disorders, therapeutic advancements

## Abstract

The intricate functionality of the vertebrate retina relies on the interplay between neurotransmitter activity and calcium (Ca^2+^) dynamics, offering important insights into developmental processes, physiological functioning, and disease progression. Neurotransmitters orchestrate cellular processes to shape the behavior of the retina under diverse circumstances. Despite research to elucidate the roles of individual neurotransmitters in the visual system, there remains a gap in our understanding of the holistic integration of their interplay with Ca^2+^ dynamics in the broader context of neuronal development, health, and disease. To address this gap, the present review explores the mechanisms used by the neurotransmitters glutamate, gamma-aminobutyric acid (GABA), glycine, dopamine, and acetylcholine (ACh) and their interplay with Ca^2+^ dynamics. This conceptual outline is intended to inform and guide future research, underpinning novel therapeutic avenues for retinal-associated disorders.

## 1. Introduction

An integral component of the central nervous system (CNS), the retina provides crucial insights into brain development, function, and disease progression. Spanning an approximate thickness of 250 µm in humans, 170 µm in rats, 165 µm in rabbits, and 160 µm in zebrafish, the retina is composed of rod and cone photoreceptors, horizontal cells, bipolar cells (BPCs), amacrine cells, glial cells, microglia, astrocytes, Müller cells, and ganglion cells [1,2,3,4,5]. Within this compact structure, these components perform visual functions, such as light reception, phototransduction, and signal processing [1]. The cellular components communicate with one another via multiple neurotransmitters, each of which is governed by calcium (Ca^2+^) dynamics distinctly to ensure proper retinal development and healthy physiological functioning. Thus, comprehension of retinal structure, its components, and its neuronal mechanisms is crucial for the advancement of research in the field.

Neurotransmission requires interaction between Ca^2+^, various neurotransmitters including glutamate, gamma-aminobutyric acid (GABA), glycine, dopamine, and acetylcholine (ACh), and their respective receptors, ensuring precise and rapid communication between retinal cells [6,7]. The heteromeric, multi-subunit neurotransmitter receptors facilitate increased diversity and specificity of their synapses [8]. In the retina, optimal neurotransmission relies on the precise interplay between neurotransmitters and Ca^2+^ dynamics, since Ca^2+^ ions act as second messengers that influence numerous cellular functions and facilitate neurotransmitter release, which is essential for maintaining visual sensitivity, perception, and signal processing [9,10,11,12]. Such interactions are present during development in retinal waves and throughout normal adult functioning with the regulation of neurotransmitter release and circadian rhythms, for instance [13,14,15,16,17,18,19]. Any disruptions of this interplay between neurotransmitter release and Ca^2+^ dynamics can lead to visual impairments and eye-related disorders, such as glaucoma, retinal ischemia, diabetic retinopathy, Parkinson’s Disease (PD) visual impairment, and form-deprivation myopia, as discussed in detail for each neurotransmitter [20,21,22,23,24,25,26,27,28].

Due to its accessibility and direct connection to the central nervous system, the study of the retina in different animal models can offer insight into the detection of neurodegenerative diseases, including Huntington’s disease (HD), PD, and Alzheimer’s disease (AD), and ocular disorders, including glaucoma, age-related macular degeneration, and diabetic retinopathy in humans [29,30,31,32]. For each of these, the pathogenicity involves changes in neurotransmission and Ca^2+^ dynamics within the retina, highlighting this unique structure as the center point of a crucial area of research for potential therapeutic intervention [20,21,22,23,24,25,26,27,28]. To date, multiple studies have identified the effects of drugs such as pilocarpine, ethosuximide, and pregabalin on Ca^2+^ regulation in treatments of various visual impairments related to the dysregulation of different neurotransmitters [33,34,35]. The current review will explore important aspects of the retina, including its structure, synapses, visual processes, and development, and describe the roles of the retinal neurotransmitters glutamate, GABA, glycine, dopamine, and acetylcholine and their interplay with Ca^2+^ ions in retinal development, normal physiological functioning, and disease.

## 2. The Retina

### 2.1. Retinal Structure and Information Processing

The complex circuitry of the retina relies on the arrangement of key components across various cellular layers that facilitate vision in various organisms. Therefore, an understanding of retinal structure is fundamental to understanding its function and mechanisms. The general structure of the human retina is shown schematically in Figure 1. Starting from the innermost layer that is closer to the anterior part of the face and the vitreous humor that fills the eye, the retina is composed of the inner limiting membrane (ILM), retinal nerve fiber layer (RNFL), retinal ganglion cell (RGC) layer, inner plexiform layer (IPL), inner nuclear layer (INL), outer plexiform layer (OPL), outer nuclear layer (ONL), outer limiting membrane (OLM), the ellipsoid zone (EZ), the photoreceptor outer segments (POSs), and the retinal pigment epithelium (RPE) [36]. As described by Masland et al. (2012), each layer is composed of different cell types with unique functions relevant to different aspects of vision [1]. The ILM, composed of astrocytes and the end feet of Müller cells, creates a barrier between the vitreous humor and the retina itself. The RNFL is formed by the axons of RGCs, the RGC layer by the cell bodies of RGCs and by displaced amacrine cells, the IPL by amacrine cells and synapses between the dendrites of RGCs and the axons of BPCs, and the INL is composed of horizontal, bipolar, amacrine, displaced ganglion, and the cell bodies of Müller cells. The OPL is formed by synapses between photoreceptors, horizontal cells, and BPCs, and the ONL is formed by the nuclei of the photoreceptors. The EZ, also known as the inner segment (IS) area, contains the inner segments of photoreceptor cells, while their outer segments reside in the POS. Finally, the RPE right behind the photoreceptors contains a simple layer of cuboidal cells and is connected to Bruch’s membrane and the choroid [1], as shown in Figure 1.

The visual process involves an interplay among various retinal cell types, each holding unique functions to contribute to the transmission of visual information. First-order neurons, the photoreceptors, are divided into two types: rods and cones. Rod photoreceptors detect low-light levels, while cone photoreceptors enable photopic vision with high spatial acuity and color differentiation [37]. Horizontal cells, which are second-order neurons, modulate information transfer between BPCs and photoreceptors and provide lateral inhibition to enhance contrast and contribute to spatial receptive fields [38]. BPCs, another type of excitatory second-order neuron, receive input from photoreceptors and provide a link between them and the different types of RGCs. The various types of BPCs play distinct roles in collecting and modulating signals from photoreceptors, initiating the processing of visual stimuli within the inner retina [39,40]. Although amacrine cells typically release inhibitory neurotransmitters, the presence of gap junctions allows them to play both inhibitory and excitatory roles, acting as interneurons that interact with cells of the vertical retinal pathway—the pathway that involves the transmission of visual information from photoreceptors to BPCs to ganglion cells—and modulate signal transmission between neurons [41]. RGCs are the main output neurons for the retina and play dual roles in image-forming and non-image-forming processes [42]. Müller glial cells are involved in various metabolic processes and provide support to retinal neurons [43].

Each neuronal class exists in specialized types to perform unique functions, with three cone photoreceptor types, approximately three different types of horizontal cells, twelve types of BPCs, approximately twenty types of amacrine cells, and up to 45 types of RGCs [44]. Although the specific classifications are out of the scope of this review, we will mention necessary classifications in different contexts, and they will be further explained within each section when needed. Here, we briefly describe the major retinal neuronal classifications. Cone photoreceptors are composed of visual pigments, called opsins, and based on the structure of their opsins, cones are sensitive to either short- (or blue), medium- (or green), or long-wavelengths (or red wavelengths) of light [45]. In this way, these neurons can be classified as short-wavelength or blue cones, medium-wavelength or green cones, and long-wavelength or red cones according to their relative spectral sensitivity [45]. There is only one type of rod photoreceptor, which contains rhodopsin, the light receptor that initiates scotopic vision [46]. Horizontal cells have three types that have been identified: HA-1, HA-2, and HB [47]. Although more research is required to determine the selectivity of horizontal cells towards different photoreceptors, HA-1 and HA-2 cells associate with cones without specificity, whereas HB cells seem to associate specifically with blue-sensitive cones given their large terminals organized in a rectilinear configuration [48]. There are, as previously mentioned, at least twelve types of BPCs, with one type connecting with rod photoreceptors (rod bipolar cell) and the remaining types connecting with cone photoreceptors (cone BPCs) [49]. These neurons are mainly divided into ON-type and OFF-type BPCs. OFF-type BPCs are hyperpolarized in the light and depolarized in the dark [50]. ON-type BPCs, on the other hand, are hyperpolarized in the dark and depolarized in the light [50]. Amacrine cell types can be broadly divided into ON-type, OFF-type, and ON–OFF amacrine cells [50]. ON-type amacrine cells receive information exclusively from ON-type BPCs, depolarizing in the presence of light. OFF-type amacrine cells receive information exclusively from OFF-type BPCs, therefore depolarizing in the absence of light. ON–OFF amacrine cells receive information from both types of BPCs and depolarize in both the presence or absence of illumination [50]. Amacrine cells may also be divided based on the neurotransmitter they release, being classified into either GABAergic amacrine cells or glycinergic amacrine cells, although a small population of these neurons has been shown to be neither GABAergic nor glycinergic (nGnG) [51]. Finally, RGCs can be classified as ON-type, OFF-type, and ON–OFF-type RGCs. ON-type RGCs are excited by illuminating their receptive field center and inhibited by illuminating their receptive field surround. OFF-type RGCs work oppositely, being excited in their receptive field surroundings and inhibited by light in their receptive field center. ON–OFF RGCs can respond to both increases and decreases in light intensity within their receptive fields [50].

These different neuronal classifications play specialized functions in visual processing. Light first enters the eye through the cornea and passes through the pupil, iris, and lens before reaching retinal photoreceptors, where phototransduction, the conversion of photons of light into electrical signals, takes place [50]. In the vertebrate retina, both rod and cones respond to light with hyperpolarizing receptor potentials [52]. In the dark, the plasma membranes of rod and cone photoreceptors in the outer segment are extremely permeable to cations, as evidenced by the so-called dark current, which is a continuous influx of sodium ions (Na^+^) into the outer segments [52]. To achieve an electrical balance, the efflux of potassium ions (K^+^) into the inner segments occurs via the activity of a Na^+^/K^+^ ATPase [53]. Photoreceptors are depolarized by the dark current, resulting in a tonic release of glutamate in the synaptic cleft that forms a signal to the postsynaptic neurons [53].

In the presence of light, absorption of photons by photo-pigments in the outer segments reduces Na^+^ conductance and decreases the dark current, resulting in hyperpolarization of the membrane potential and a decrease in the release of tonic glutamate [54]. The sign-inversing and sign-conserving responses of bipolar neurons to glutamate are discussed in Section 3.1.2. Briefly, under light stimulation, ON-center BPCs become depolarized, while OFF-center BPCs become hyperpolarized. BPCs release glutamate into the synaptic cleft to communicate with RGCs and amacrine cells in the IPL [55]. Depending on whether the signals are from ON or OFF BPCs, RGCs convey action potentials to the visual cortex for further processing [56]. Horizontal cells also hyperpolarize in response to light, thereby reducing the release of the neurotransmitter GABA in synapses with cone photoreceptors and BPCs [55]. Signals from neighboring rod and cone photoreceptors converge onto the same RGC. Such convergence allows rod photoreceptors to combine smaller signals and produce more substantial responses in BPCs [57]. Finally, the resulting signals are transmitted to the brain through the RGCs, which form the optic nerve through the combination of their axons [50].

This review will focus on information from the literature on mouse and primate retinas. The mouse retina is rod-dominated with a rod–cone ratio of 35:1 [58,59]. Only 5% of cones express purely a short (S) wavelength-sensitive opsin, while the remaining co-express S-opsin and a middle (M) wavelength-sensitive opsin [60,61]. On the other hand, the primate retina is enriched with cone photoreceptors with a rod–cone ratio of 20:1 [62]. Three opsins are expressed by the correspondent types of cones: S (blue), M (green), and L (red), allowing for fine discrimination between colors in the visible spectrum. The central retina of primates also contains a region specialized for high acuity called the fovea. Within the cone-abundant fovea, a single cone photoreceptor contacts a synapse with a single (midget) bipolar cell, which in turn contacts a single (midget) ganglion cell [63], ensuring the highest acuity vision by providing each cone a private line of communication to the brain [64]. The arrangement of photoreceptors in the primate retina is matched to the function of the foveal vs. the peripheral retina. The foveal region is crucial for color vision, with a ratio of cone types of L:M:S cones of 11:5:1 [65], whereas the peripheral retina is more important for luminance coding and, therefore, abundant in rod photoreceptors.

### 2.2. Retinal Synapses

As we introduced in the previous section, neurons must communicate with each other precisely to transmit signals from photoreceptors to BPCs via synaptic transmission. Retinal synapses are composed of two different types: conventional synapses, and specialized ribbon synapses. Within the CNS, the most common type of synapse is the conventional (also called central) synapse, which consists of an active zone containing synaptic vesicles in the pre-synaptic terminal and a post-synaptic density in the post-synaptic terminal. Conventional synapses are formed by all retinal cell types, except for photoreceptors and BPCs, which form ribbon synapses and vary in their functional properties based on cell type [66].

Systems such as vision and hearing present ribbon synapses, specialized to respond to sensory stimuli and transmit these signals to appropriate cells to encode diverse information via neurotransmission [67]. The active zone of these synapses contains dense ribbon-like specialized organelles, called synaptic ribbons, that are tethered to the cell membrane [68]. In the retina, photoreceptors and BPCs contain such ribbon synapses, signaling graded changes in membrane potential [11,68,69,70]. Ribbons tether synaptic vesicles and calcium channels and have been hypothesized to play important roles in the facilitation of synaptic vesicle release within the synaptic vesicle cycle in a calcium-dependent manner, with vesicles being released when calcium channels are opened [69,71,72,73,74,75,76,77]. In ribbon synapses, Ca^2+^ influx has been shown to occur in hotspots located near the ribbon through voltage-gated Ca^2+^ (Ca_V_) channels [69,77].

Ca_v_ channels are grouped into Ca_V_1, Ca_V_2, or Ca_V_3 subtypes based on the properties of the currents that the channels mediate [78]. Ca_V_1 channels mediate L-type currents, which are long-lasting currents that require strong polarization for their activation [79]. Ca_V_2 channels mediate P/Q-, N-, and R-type currents, which are activated by high voltage and classified based on their sensitivity to different blockers and toxins [80,81]. Ca_V_3 channels mediate T-type currents, which are transient currents that are activated by low voltage [79,80]. Ribbon synapses are driven by L-type Ca channels, or Ca_V_1 channels, which are further classified into Ca_V_1.1, Ca_V_1.2, Ca_V_1.3, and Ca_V_1.4 [78]. These channels are tightly coupled to “big-potassium”, or BK, channels at presynaptic active zones in order to regulate the entry of calcium and the release of neurotransmitters [82]. BK channels are voltage-gated potassium channels that have a large K^+^ conductance across the cell membrane when activated by membrane depolarization and Ca^2+^ entry through Ca_V_ channels [83].

In conventional synapses, BK channels have been described as negative regulators of neurotransmitter release in frogs and lizards [84,85]. Similarly, in conventional synapses in the retina of rats, the contribution of Ca_V_1 channels of amacrine cells to GABA release was reduced by BK channels as they suppressed postsynaptic depolarization of the neurons and limited the activation of Ca_V_ channels [86]. In ribbon synapses, on the other hand, results have shown mixed and dual roles of BK channels in neurotransmission. In the rod photoreceptors of salamanders, BK channels were found to facilitate transmitter release, with BK channel activation increasing extracellular levels of K^+^, therefore enhancing Ca^2+^ channel currents and amplifying synaptic transmission [87]. In the mouse retina, BK channels did not seem to modulate the activity of cone BPCs and photoreceptors but modulated the activity of rod BPCs instead, where the absence of BK channels under scotopic light conditions results in a significant reduction in electroretinogram signals [88]. Therefore, the function of BK channels and their interactions with Ca_V_ channels seems to vary based on experimental and physiological conditions, requiring a clearer understanding of the variability among studies and differences between the activity of Ca^2+^ and K^+^ channels in different retinal neuronal types.

### 2.3. Retinal Development

Retinal development is a tightly regulated process that occurs during embryonic and early postnatal time. Across species, the basic components of the eyes and retina retain similar functions, although the timing of critical developmental events varies from one species to another [89]. This review will focus on vertebrate development, with a focus on mice, as retinal development in this species has been well-detailed by Hoon et al., 2014 and Cepko, 2014 [90,91]. Figure 2 illustrates the timescale of embryonic and postnatal development for various retinal neuronal types in mice. This figure will be recurrent in this paper as we detail the activity and presence of different neurotransmitters during retinal development [90,91].

Similar to other regions of the CNS, the developing retina is derived from the neural tube and is formed by the invagination of the optic vesicle to form the optic cup [92]. The external part of the optic cup becomes the RPE, while the internal part develops into the multi-layered retina [93]. The RGCs are the first cells to differentiate, at approximately Embryonic day 10 (E10) in mice, and extend their axons to form the optic nerve [94]. As development progresses, other cell types differentiate and establish connections within the retina in the following order: horizontal cells, cone photoreceptors, amacrine cells, rod photoreceptors, BPCs, and finally, Müller glial cells [94,95].

Throughout the development of the retina, events like cell migration, fate determination, and synaptogenesis ensure proper retinal organization and function [96,97,98,99]. After birth, the retina continues to refine and mature via visual experience and synaptic plasticity, with synaptic connections being finely tuned for proper visual pathways and receptive fields [100]. During retinal development, neurotransmitters play vital roles in shaping neural circuits and establishing functional connections, thereby contributing to the precise processing and transmission of visual information in the mature retina. The refinement of retinal connections and receptive fields relies on Ca^2+^-dependent and Ca^2+^-independent processes together with a fine balance between excitatory and inhibitory neurotransmitters, ensuring the development of a proper visual pathway. The culmination of retinal development is a well-organized and functional neural network that facilitates the detection and transmission of visual information to the brain.

## 3. Neurotransmission

This section will provide an overview of the current understanding of the retinal neurotransmitters glutamate, GABA, glycine, dopamine, and ACh and of their interplay with Ca^2+^ in multiple retinal mechanisms. Table 1 summarizes the roles of these neurotransmitters in retinal development and normal retinal physiology, Table 2 summarizes their involvement in different retinal diseases, and Table 3 outlines their interplay with Ca^2+^-dependent processes discussed herein.

### 3.1. The Roles of Glutamate in Retinal Development, Normal Physiology, and Disease and Its Interplay with Calcium

The primary excitatory neurotransmitter in the nervous system is the amino acid glutamic acid (glutamate), which plays a key role in human retinal function. Glutamate is released by photoreceptors, bipolar neurons, vglut3 amacrine cells, and RGCs. It is primarily involved in the vertical pathways of the retina [140,141], where it stimulates postsynaptic membranes via rapid depolarization and Ca^2+^ ion influx [14,134]. Glutamate receptors are classified into ionotropic receptors (iGluRs), which consist of NMDA receptors, AMPA receptors, kainite receptors (KARs), and metabotropic receptors (mGluRs). Ionotropic receptors are usually ligand-gated ion channels, whereas metabotropic receptors activate signaling cascades through G-proteins and second messengers [142].

#### 3.1.1. The Role of Glutamate in Retinal Development

Glutamate regulates spontaneous neural activity during retinal waves and guides activity-dependent processing of the visual field map in RGCs, being crucial in their development and maturation [18,101,102]. Retinal waves are composed of spontaneous neural activity that occurs during RGC development and serve as a foundation for the activity-dependent refinement of retinotopic maps in the brain [143]. While the establishment of glutamatergic retinal waves was initially attributed to amacrine cells and BPCs, more recent reports reveal the involvement of Müller glial cells in the process [19]. During the occurrence of retinal waves, the ionotropic α-amino-3-hydroxy-5-methyl-4-isoxazolepropionic acid (AMPA) glutamate receptors regulate Ca^2+^ activity in Müller glial cells, while the glutamate transporters, including the excitatory amino acid transporters EAAT_1_ and EAAT_2_, play a modulatory role, allowing the cells to sense and control the retinal waves by adjusting the concentration of extracellular glutamate [19,144]. In mice, glutamatergic retinal waves occur between postnatal days 10 and 14 [16,126], as shown by the grey-shaded bar in Figure 3.

The retina also expresses Ca^2+^-permeable N-methyl-D-aspartate (NMDA) receptors for glutamate that, together with non-NMDA receptors, regulate the outgrowth and stabilization of RGC dendrites during pre-embryonic development and their remodeling during further postnatal development [102,103,104]. In many experimental animal models, neurons in the IPL and GCL express high levels of NMDA receptors, beginning during the early phases of retinal development and increasing after birth. These results suggest that they are likely involved in early retinal development and the establishment of synapses within the inner retina, primarily in amacrine cells and RGCs [103,105,145,146]. Thus, NMDA receptors likely regulate the differentiation of different cell types during pre-embryonic development [103].

After birth, glutamate-mediated neuronal activity may be more prominent in defining the structure of the inner retina, primarily the remodeling of the dendrites and stratification of RGCs [101,103]. NMDA conductance determines the amount of current that can pass through NMDA receptor channels when open. Interestingly, NMDA conductance decreases as retinal development progresses, meaning that the receptor channels become less effective [104]. In contrast, the rate of desensitization of ionotropic glutamate receptors increases as development progresses [104]. It is hypothesized that such a change in behavior throughout the developmental process likely enables RGCs to integrate synaptic information, facilitating the transmission of electrical signals to the brain. These observations reinforce the idea of the various roles played by glutamate throughout retinal development, although more research will be required to form a more complete understanding of the functions of glutamate and its receptors during development.

#### 3.1.2. The Role of Glutamate in Normal Retinal Physiology

In the mature retina, glutamate acts on vertical pathway neurons and is used primarily by photoreceptors as an excitatory neurotransmitter for the relay of signals to second-order neurons [55]. While photoreceptors release glutamate upon depolarization, providing the basis for signal transmission and subsequent visual processing [110], the mechanism of release is distinct for cone and rod photoreceptors. Cones release glutamate solely through synaptic ribbon sites, whereas, similarly to BPCs, rods exhibit a broader release pattern from both ribbon and non-ribbon sites [147,148,149]. Glutamate undergoes modifications that are mediated by horizontal cells and different bipolar cell types, and is further refined by amacrine cells [50]. ON BPCs express metabotropic glutamate receptors and exhibit sign-inversing responses by coupling to G-proteins, in which glutamate hyperpolarizes the membrane of ON BPCs, inhibiting synaptic transduction currents. Glutamate receptors of OFF BPCs, on the other hand, display sign-conserving behavior, in which an increase in glutamate signaling results in an increased electrical response of the OFF BPCs [106,107,108].

The distribution and activity of glutamate receptors further contribute to their role in normal retinal functioning. Glutamate receptor subtypes, including AMPA, NMDA, and KA receptors, are differentially located across different neurons (Figure 4), where they play distinct roles. The AMPA and KA receptors are located at postsynaptic density sites, where they allow for the passage of primarily Na^+^ and K^+^ ions in OFF BPCs, horizontal cells, and RGCs, while the peri-synaptic NMDA receptors are present in some ON BPCs, amacrine cells and RGCs, and permeable to Na^+^, K^+^, and Ca^2+^ ions [109,110,111]. NMDA receptors, alongside AMPA receptors, are capable of modulating glutamate release in the retina without significant limitation of the temporal tuning of synaptic responses in RGCs [111].

Alongside iGluRs, the retina expresses mGluRs, which are present in eight types across the different retinal neurons (mGluR1-8). In rodents, mGluR1 has been shown to be present in bipolar, amacrine, and RGCs; mGluR2 in amacrine and RGCs; mGluR3 in RGCs; mGluR4 in amacrine and RGCs; mGluR5 in horizontal, bipolar, amacrine, and RGCs; mGluR6 in bipolar and RGCs; mGluR7 in bipolar, amacrine, and RGCs; and mGluR8 in photoreceptors, horizontal, amacrine, and RGCs (Figure 4) [150]. The sign-inversing response of the metabotropic glutamate receptors described earlier is possible especially because of mGluR6 on the dendrites of ON BPCs. mGluR6 binds the glutamate from photoreceptors and activates the G protein G_o_, which leads to the closing of the cation channel TRPM1 (Transient Receptor Potential Cation Channel Subfamily M Member 1), therefore hyperpolarizing the neuron [151,152,153].

#### 3.1.3. The Role of Glutamate in Retinal Diseases

Dysregulation in the balance of glutamate can impact retinal function. The metabolism of glutamate is governed by its uptake and conversion by adenosine triphosphate (ATP)-dependent glutamine synthetase (GS) to non-toxic glutamine in Müller glial cells, which is a highly energy-dependent process [154]. Thus, a lack of energy can disrupt the cell’s glutamate equilibrium, resulting in oxidative stress, mitochondrial impairment, and upregulation of NMDA receptors, ultimately inducing excitotoxicity, which is the damage caused to neurons by excessive stimulation from neurotransmitters like glutamate [155,156]. This process has been implicated in the degeneration of RGCs, which vary in sensitivity by cell type, with α-type RGCs, which are RGCs with large somata and dendritic field [157], displaying more resistance and J-type RGCs, which are direction-selective cells that respond specifically to upward motion [158], being the most sensitive [44]. Excitotoxicity can also result from excessive stimulation of glutamate receptors, underlying a pivotal mechanism that contributes to retinal pathology [159,160]. Interestingly, this process can be mitigated by glutamate-release inhibitors, which may serve as the basis for potential therapeutic interventions that aim to reduce glutamate neurotransmitter activity [161].

Dysfunctional glutamate metabolism is linked to retinopathies such as retinal ischemia, which is induced by the occlusion of retinal or choroidal blood vessels. Acute retinal ischemia sparks a critical depletion of ATP and glutamine synthetase [162,163,164], resulting in elevated levels of extracellular glutamate that trigger a destructive excitotoxicity cascade characterized by neuron depolarization, the influx of Ca^2+^ ions, oxidative stress, and robust glutamatergic stimulation, which contributes to the degenerative repercussions of ischemia and reperfusion scenarios [165]. Notably, the function of Müller cell glutamate aspartate transporter (GLAST) decreases significantly after retinal ischemia, resulting in an elevation of glutamate levels within the aqueous humor and the inner retina [166,167,168]. However, this impairment is not permanent, since the function of GLAST is recovered on reperfusion [169].

The pathogenesis of the ocular disorder glaucoma, characterized by the apoptotic loss of RGCs, remains poorly understood but appears to involve glutamate excitotoxicity [131]. A reduction in the levels of the glutamate transporter GLAST is associated with increased intraocular pressure (IOP), poses a significant risk for the development and progression of glaucoma, causes damage to the optic nerve, and ultimately leads to vision loss [133,170]. Interestingly, in animal models of glaucoma, the absence or downregulation of different glutamate transporters resulted in the death of RGCs, even in the absence of high IOP [170]. In normotensive glaucoma, deletion of the apoptosis signal-regulating kinase 1 (*ASK1*) gene suppresses the oxidative stress that is linked to glutamate-mediated excitotoxicity, increases survival of RGCs, and reduces the production of the pro-inflammatory cytokine tumor necrosis factor-alpha (TNF-α) by macrophages, microglia, and astrocytes [132,171]. Thus, a potential therapeutic approach to reverse the effects of glutamate-related toxicity might include *ASK1* inhibitors, such as the highly selective inhibitor selonsertib.

The progression of retinal degeneration in diabetic retinopathy, once thought to result from a microcirculatory disorder, is now known to involve glutamate and Ca^2+^ signaling pathways. The early stages of diabetes mellitus are associated with heightened levels of glutamate and oxidative stress [172,173]. In rat Müller cells, NMDA receptors inhibit the secretion of vascular endothelial growth factor (VEGF), thereby countering blood–retinal barrier breakdown and indicating the protective effect of a well-maintained glutamatergic system [174,175]. In rat models of diabetes, the NMDA receptor inhibitor memantine blocks elevated levels of VEGF [175]. However, as noted above, the protective role of glutamate is compromised in diabetic retinopathy, since Müller cells in the inner retina of diabetic individuals exhibit decreases in GLAST activity and Müller cell immunoreactivity [176,177]. To mitigate the impact of dysregulated glutamate and Ca^2+^ signaling in diabetic retinopathy, novel therapeutic targets might include the mitochondrial membrane lipid cardiolipin, which is linked to apoptosis cascades, and antioxidant factors such as lipoic acid, carotenoids, and nuclear factor (erythroid-derived 2)-like 2 (Nrf2), which plays a crucial protective role against oxidative stress [178,179,180,181,182].

#### 3.1.4. The Interplay between Glutamate and Calcium

As described above, retinal waves generate spontaneous bursts of Ca^2+^ that occur during retinal development [126]. During retinal waves, depolarization of ON cone BPCs leads to a transient increase in intracellular Ca^2+^ levels and a release of glutamate to the next neuronal cells, the ON RGCs, which generate bursts of action potentials in the form of retinal waves [183]. In the presence of glutamate, RGCs and amacrine cells display a significant increase in Ca^2+^ levels [184]. These responses are initially mediated by AMPA and KA receptors, but the NMDA receptors fulfill this role as development progresses, underlining the hypothesis that the different types of receptors play different roles during development. The interplay between glutamate activity and changes in intracellular Ca^2+^ levels is crucial for the outgrowth, stabilization, and differentiation of various cell types, especially amacrine cells, and RGCs, during development [102,104,145,184].

In normal physiological conditions, glutamate is present in vast amounts, especially at retinal ribbon synapses where synaptic vesicles filled with glutamate are attached [11]. Ribbons are crucial in the pre-synaptic release of glutamate, in which they assist with the replenishment of synaptic vesicles and organize the proteins necessary for glutamate release, vesicular clustering, and positioning of voltage-gated Ca^2+^ channels at synaptic active sites [185,186]. Non-ribbon glutamate release also occurs in the retina, involving Ca^2+^-mediated processes, such as Ca^2+^-induced calcium release (CICR) from intracellular endoplasmic reticulum (ER) stores [187,188].

In retinal ischemia, the rise in intracellular Ca^2+^ levels determines the beginning of excitotoxic glutamate-related neuronal death [135] that occurs in RGCs of rats due to the activation of AMPA/kainite receptors during visual processing, which facilitates the influx of Ca^2+^ through NMDA receptors [134]. In living animals, such depolarization of RGCs is mediated by glutamate release during visual stimuli in the retina [134]. One possible mechanism of cell death in diabetic retinopathy involves a step-wise increase in retinal glutamate levels, activation of NMDA receptors, increased levels of intracellular Ca^2+^, induction of RGC apoptosis, and further activation of NMDA receptors, a cycle that contributes to the death of numerous retinal cells [136].

**Table 3 ijms-25-02226-t003:** Summary of calcium interplay with different neurotransmitters in retinal development, health, and disease, as discussed in the present review.

Neurotransmitter	Role of Calcium Interplay
Glutamate	Triggers retinal waves during development [183]. Crucial for the outgrowth, stabilization, and differentiation of amacrine cells and RGCs during development [102,104,145,184]. Regulate replenishment of synaptic vesicles at synaptic active sites [185,186]. Initiates glutamate-related excitotoxicity in retinal ischemia [135]. Implicated in glutamate-related diabetic retinopathy and RGC death [136].
GABA	GABA regulates retinal Ca^2+^ waves during development via GABAB receptors in amacrine cells and RGCs [189]. Reduced Ca^2+^ levels are associated with reduced GABA release in pre-synaptic GABAergic amacrine cells in diabetic retinopathy [190]. Some antiepileptic drugs reduce Ca^2+^ currents and intracellular Ca^2+^ levels and calcium currents [34].
Glycine	Inhibits voltage-gated Ca^2+^ currents through a G-protein pathway, reducing signal transmission from BPCs to RGCs [191]. ATP suppresses glycine receptor mediated Ca^2+^ currents in OFF RGCs, altering the pattern of action potentials in the retina [192]. Glycine is potentiated by cyclic AMP (cAMP) through EPAC2 and Ca^2+^ stores in AII-type amacrine cells. EPAC2 activation results in Ca^2+^ release from Ca^2+^ stores, resulting in an increase of glycine release onto OFF-cone bipolar cell terminals [193].
Dopamine	D4 receptors in photoreceptors modulate Ca^2+^ currents [194,195]. Ca^2+^-dependent balance between melatonin and dopamine release [196]. Calcium dynamics moderate retinal processes that regulate the circadian rhythm [197].
Acetylcholine (ACh)	Propagates Ca^2+^ bursts during retinal development [198,199]. G protein-coupled mAChRs activate the alpha subunits of Gq/11 proteins, raising the levels of intracellular Ca^2+^. This Ca^2+^ increase enhances PKC activity and may stimulate NO production via Ca^2+^-dependent neuronal nitric oxide (NO) synthase [200]. M2 and M4 receptors inhibit the cAMP-dependent pathway, indirectly influencing Ca^2+^ dynamics [200]. Associated with neuroprotective roles, preserving Ca^2+^ balance and mitochondrial membrane integrity with drugs such as pilocarpine, which is effective against glaucoma [35].

### 3.2. The Roles of Gamma-Aminobutyric Acid (GABA) in Retinal Development, Normal Physiology, and Disease and Its Interplay with Calcium

The principal inhibitory neurotransmitter in the CNS and retina is gamma-aminobutyric acid (GABA), which plays a critical role in regulating visual processing. GABA is released, primarily via Ca^2+^-dependent conventional exocytosis and Ca^2+^-independent mechanisms via GABA transporters [201], from GABAergic horizontal cells, amacrine cells, to BPCs and RGCs, where GABA is crucial for the processing of visual information and the development of retinal circuitry in the inner retina [202].

#### 3.2.1. The Role of GABA in Retinal Development

As noted above, GABA released by amacrine cells and RGCs is an important regulator of inner retinal development [7,203,204,205,206,207,208]. GABA_B_ receptors are metabotropic G protein-coupled receptors on amacrine cells and RGCs, where they are capable of inducing neuronal hyperpolarization through the opening of G protein-gated inwardly rectifying K^+^ channels and of inhibiting neurotransmitter release by inhibiting voltage-gated Ca^2+^ channels both pre- and post-synaptically [115]. During early neurogenesis in the retina of embryonic chickens, an agonist of GABA_B_ blocked Ca^2+^ signaling during retinal Ca^2+^ waves, while the GABA_B_ antagonist p-3-aminopropyl-p-dietoxymethyl phosphoric acid increased the duration of the transient, illustrating that the GABA receptor subtype plays a crucial role in regulating Ca^2+^ waves prior to synapse formation [112]. In the mouse retina, expression of the enzyme L-glutamate decarboxylase (GAD) was used to track the developmental pattern of GABAergic neurons (Figure 5). GAD expression was robust during the period that spanned embryonic day 17 to postnatal day 3 and then significantly declined after postnatal day 12 [209].

GABA is also implicated in postnatal stages of retinal development. During the early postnatal weeks in the ferret retina, GABA potentiates activity on all RGCs through the action of GABA_A_ receptors [113]. As development progresses and ON–OFF cell segregation occurs, the role of GABA shifts to the suppression of ganglion cell burst activity [113]. This developmental transition underscores the multifaceted roles of GABA in shaping distinct patterns of presynaptic activity, extending its functions its inhibitory characteristics.

#### 3.2.2. The Role of GABA in Normal Retinal Physiology

GABA plays an important role in shaping the retinal response to visual stimuli. Its physiological impact is mediated via three types of membrane receptors: ionotropic GABA_A_ and GABA_C_ receptors and metabotropic GABA_B_ receptors [210,211,212]. The GABA_A_ receptor is organized in groups of five subunits, out of seven existing subunits (α, β, γ, δ, ε, π, and θ), that encircle a chloride channel and are highly sensitive to the receptor antagonist bicuculline and the chloride channel blocker picrotoxin [213,214]. There are multiple isoforms of these subunits, with six *α* (1-6), four *β* (1-4), three *γ* (1-3), and one *δ* that have been cloned [215]. Benzodiazepines are agonist drugs that allosterically bind to GABA_A_ receptor *α* and *γ* isoforms, increasing the receptor sensitivity to GABA, which in turn increases the opening frequency of Cl^−^ channels [216,217,218]. Interestingly, the α_4_ and α_6_ subunit isoforms are not sensitive to benzodiazepines given their differential peptide sequence containing an arginine instead of a histidine that prevents binding [219,220]. In addition, the *α*_2_ subunit is crucial for direction-selective inhibition in the retina, as a GABA_A_ receptor *α*_2_ mice knockout model resulted in significant impairment of direction-selective responses in calcium imaging recordings of RGCs and SACs [221]. GABA_A_ receptors have been shown to be present in photoreceptors, horizontal cells, bipolar cell axons and dendrites, amacrine cell processes, RGCs, and Müller glial cells in different species [203,204]. As demonstrated in mice, GABA_A_ receptor activation inhibits the proliferation and self-renewal of retinal progenitor cells [114]. Metabotropic G protein-coupled GABA_B_ receptors in the retina are present in horizontal neurons, amacrine neurons, and RGCs [222], playing crucial roles in signal modulation, inhibiting neurotransmitter release from the presynaptic terminal, or modulating the balance of brief and sustained signals in the retinal circuitry [115]. GABA_C_ receptors are present at different synaptic sites in cone photoreceptors, horizontal cells, BPCs, amacrine cells, RGCs, and gate chloride ion (Cl^−^) channels [110]. They function as auto-receptors, limiting the release of GABA itself [223]. GABA_C_ receptors have been identified as GABA receptors that are resistant to bicuculline, and baclofen, a GABA_B_ receptor agonist [116]. However, given that the retinal bipolar neurons of goldfish contain two distinct bicuculline/baclofen-resistant receptors, one coupled to Cl^−^ channels (a GABA_A_-type receptor) and another coupled to Ca^2+^ channels through a GTP-dependent process (a GABA_B_-type receptor), Matthews et al. proposed that receptors that were previously classified as GABA_C_ receptors belong in the category of GABA_A_ and GABA_B_ receptors instead [116].

The release of GABA is tightly regulated by both Ca^2+^-dependent and Ca^2+^-independent mechanisms [224]. In the catfish retina, an increase in the concentration of intracellular Na^+^ triggers GABA transporters to release GABA via a non-vesicular, Ca^2+^-independent mechanism [224]. Ca^2+^-dependent mechanisms occur upon the influx of Ca^2+^ ions during depolarization, ultimately leading to the release of different neurotransmitters. In such mechanisms, for example, during brief depolarization in retinal amacrine cells, the intracellular release of Ca^2+^ from internal stores further elevates its concentration, resulting in GABA release [225]. Following the release of GABA, its concentration is maintained collectively by the GABA transporters (GAT1, GAT2, GAT3, and GAT4) and the vesicular transporter VGAT [226,227,228], while the termination of the signal is achieved via uptake mechanisms that involve both presynaptic terminals and glial cells [229]. In the inner retina, amacrine cells and Müller glial cells participate in GABA uptake, while Müller glial cells are the neurons that are primarily responsible for this function in the outer retina [230]. Upon entry into Müller cells, GABA is metabolized by the mitochondrial enzyme GABA transaminase to yield succinate semialdehyde [230]. Figure 6 illustrates the localization of the different types of GABA receptors in the retina.

#### 3.2.3. The Role of GABA in Retinal Diseases

GABA dysregulation is associated with diabetic retinopathy and retinal ischemia. Diabetic retinopathy involves alterations in GABA metabolism in Müller cells whereby heightened activity of GAD leads to an increase in the accumulation of GABA while the [22] GABA transaminase (GABA-T) that metabolizes GABA exhibits reduced activity [22]. In rats, this shift is accompanied by the detection of reduced oscillatory potentials (OPs) using an electroretinogram (ERG), highlighting the functional impairment associated with diabetic retinopathy [22]. Retinal ischemia, a pathological condition previously discussed in terms of glutamate activity, leads to the rapid accumulation of GABA in a manner similar to that observed for glutamate [23]. Interestingly, in glial cells, the accumulation of GABA is more rapid and reaches higher peak levels due to a decrease in GABA transaminase activity in these cells [231]. Thus, glial cells will be relevant for the development of novel targeted treatments for retinal ischemia.

Therapeutics that target GABA pathways have the potential for the treatment of various diseases that affect retinal health. For example, anticonvulsants that influence GABAergic signaling also exert neuroprotection of RGCs against excitotoxicity, thereby representing promising treatments for retinal ischemia [25]. The GABA-transaminase inhibitor vigabatrin (VGB) shows efficacy against retinal ischemia, although with side effects that include possible irreversible visual field defects [232]. Similarly, an inhibitor of the high-affinity GABA transporter GAT1 tiagabine (TGB) offers reversible protection of the retina but may cause altered color perception [233]. Recent investigations involve the use of chloride derivatives as alternative therapeutics that maximize positive outcomes while minimizing irreversible side effects, thereby bypassing retinal risks [234].

#### 3.2.4. The Interplay between GABA and Calcium

The interplay between GABA and Ca^2+^ during retinal development occurs in various animal models. In the turtle retina, GABA is responsible for changes in the dynamics of retinal Ca^2+^ waves throughout development [189]. Similarly, GABA_B_ receptors in chicken retinas regulate the retinal Ca^2+^ waves that appear early in development [112]. GABA_B_ receptors appear to modulate the rate and duration of Ca^2+^ waves in the retina by inhibiting Ca^2+^ channels, resulting in a decrease in the influx of Ca^2+^ in RGCs and amacrine cells [112]. The release of GABA from amacrine cells is dependent on a slow, asynchronous release of Ca^2+^ resulting from its accumulation in the cell terminals [235]. Horizontal cells also release GABA to photoreceptors, where it stimulates the opening of Ca^2+^ channels, leading to the depolarization of photoreceptors to balance the hyperpolarization caused by light [236,237].

The levels of Ca^2+^ are also related to diseases in which GABA is involved. For example, in diabetic retinopathy, reduced Ca^2+^ levels in pre-synaptic GABAergic amacrine cells are associated with a reduced release of GABA [190]. While little is known about the effects on Ca^2+^-dependent mechanisms of the antiepileptic drugs used to treat retinal ischemia, other drugs typically used to treat epilepsy such as ethosuximide are associated with reduced Ca^2+^ currents through T-type channels [34]. Additionally, intraocular pressure (IOP) in the eye can damage the optic nerve, contributing to the risk of developing glaucoma. Chintalapudi et al. (2017) used mice to test the effects on IOP of the gabapentinoid drug pregabalin, which is specific for the Ca^2+^ voltage-gated channel auxiliary subunit alpha2delta1 (*Cacn2d1*) gene [33]. These authors found that pregabalin reduces the flow of Ca^2+^ through its selective channels, thereby reducing intracellular Ca^2+^ levels and lowering the impact of IOP [33].

### 3.3. The Roles of Glycine in Retinal Development, Normal Physiology, and Disease and Its Interplay with Calcium

The amino acid neurotransmitter glycine is found in the synapses between AII amacrine cells and OFF RGCs, where it plays an integral part in scotopic (low-light) vision [238]. As one of the main inhibitory amino acids in the CNS, glycine helps to shape neuronal pathways and facilitates visual processing in the dark [121].

#### 3.3.1. The Role of Glycine in Retinal Development

Glycine receptors exhibit both excitatory and modulatory properties that are crucial for the development of retinal synapses. Neonatal glycine receptors (GlyRs), composed of α2 subunit homopentamers, contribute to the establishment of a depolarizing glycine-gated Cl^−^ flux that subsequently triggers Ca^2+^ influx. This process is integral to neuronal specialization, including the development of glycinergic synapses [117,118]. GlyR α2 subunits are expressed at very high levels in retinal progenitor cells of both the developing and the adult retina (Figure 7), starting at birth (postnatal day 0) in mice [239]. The accumulation of GlyR α1, α3, α4, and β subunits also exhibits a steady increase during postnatal retinal development [240]. However, there is a paucity of studies regarding the role of glycine in retinal development, and further research will be necessary to uncover its mechanisms and influences during development.

#### 3.3.2. The Role of Glycine in Normal Retinal Physiology

In the retina, glycine is found in some cone BPCs and most amacrine cells, the latter of which regulate inhibitory neurotransmission [119] as follows. The retina segregates visual signals into ON and OFF pathways before they ascend to the visual cortex. Notably, AII (rod) amacrine cells release glycine at their synapses with OFF BPCs, other amacrine cells, and RGCs, parsing out signals from the rod photoreceptors to both the ON and OFF pathways in the INL—with transmission of signals from AII to the ON pathway relying on gap junctions instead of glycine. [241]. Glycine and glutamate play vital roles in modulating crossover pathways that lead to the dominant ON crossover inhibition to OFF BPCs and RGCs, as shown in Figure 8, which was adapted from Hsueh et al., 2008. OFF RGCs receive glutamatergic OFF excitation from OFF BPCs, which is compensated by crossover inhibition of OFF BPCs and RGCs from ON amacrine cells [120]. This crossover inhibition is compensated by glutamatergic excitation of ON amacrine cells from ON BPCs [120]. Finally, crossover inhibition from OFF amacrine cells onto ON amacrine cells compensates for the previously mentioned glutamatergic excitation [120].

Post-development, the functional roles of glycine are facilitated by glycine receptors (GlyRs) composed of α and β subunit heterooligomers [242]. The GlyRs are ligand-gated ion channels that are sensitive to strychnine, which increases the permeability of the postsynaptic membrane to chloride ions [243]. Their distribution spans the IPL, INL, and RGC layer, with different kinetic speeds that are tailored to their functions. For example, in BPCs and A-type RGCs, α1β-containing GlyRs facilitate rapid signal transmission, with spontaneous inhibitory synaptic currents (sIPSCs) displaying medium fast kinetics with a lower decay time constant, while in AII amacrine cells, α3β-containing GlyRs relay rod light signals with reduced temporal resolution [119]. In narrow-field amacrine cells, α2β- and α4β-containing GlyRs contribute to modulatory functions where temporal precision is less important [119]. Notably, the α1 subunits of GlyRs play a critical role in scotopic vision, particularly at the synapses of AII amacrine cells and OFF RGCs, where they modulate the transmission of visual signals under low-light conditions [121].

Following the release of glycine and binding to glycinergic receptors, the glycine transporters GlyT1 and GlyT2 remove the neurotransmitter from the extracellular space [244]. The transporters differ in the Na^+^/Cl^−^ gradients they use in the transport of glycine, with GlyT1 using 2 Na^+^ per Cl^−^ to remove one molecule of glycine and GlyT2 using 3 Na^+^ per Cl^−^ [245]. They are also expressed in different retinal cells, although such expression varies across species. GlyT1 is predominantly found in Müller glial cells in amphibians, but it is found in amacrine cells and processes in the IPL in mammalian and chick retinas [246,247,248]. GlyT2 is present in the soma of amacrine cells and processes in the IPL and OPL of amphibians but is absent in the mammalian retina [246,247,248]. Although more research is needed to establish the function of the transporters in the retina of different species, in the mouse retina, GlyT1, in addition to playing a crucial role in the regulation of intracellular glycine concentration and maintaining the transmitter’s phenotype of AII amacrine cells, is crucial for the replenishment of the presynaptic glycine pools in these cells, a function previously thought to be unique to GlyT2 given its role in the brain stem and spinal cord [244,249,250].

#### 3.3.3. The Role of Glycine in Retinal Diseases

Since diabetes mellitus is characterized by a significant reduction in the levels of the hormone insulin, which is crucial for amino acid metabolism, it has been hypothesized that the pathogenesis of the disease is related to a deficiency of amino acids, including glycine [26]. For this reason, the therapeutic potential of glycine supplementation has been explored for the treatment of diabetic retinopathy. In diabetic rats, treatment with glycine improved the chromatin distribution and nuclear profiles of RGCs facing apoptosis [28]. Glycine possesses antioxidant, anti-inflammatory, and anti-glycation properties [251]. On glycine treatment, Müller cells exhibit increased expression of glial fibrillary acidic protein (GFAP) and other intermediate filament proteins, resulting in the accumulation of advanced glycation end products (AGEs) and inflammatory mediators and an increase in the levels of oxidative stress [251]. Glycine also prevents Müller cell reactivation and improves disorganized arrangements of the outer segments of cone and rod photoreceptors [251]. Finally, glycine supplementation attenuates retinal neuronal damage in rat models of diabetes, ameliorating its detrimental effects on the INL, ONL, and RGCL [252].

#### 3.3.4. The Interplay between Glycine and Calcium

While the interplay between glycine and Ca^2+^ during development or in retinal diseases remains largely unknown, it is observed during normal physiological function. Glycine signaling via a G-protein-associated receptor pathway inhibits voltage-gated Ca^2+^ currents, thereby reducing signal transmission in the retina from BPCs to RGCs [191]. Furthermore, ATP suppresses the Ca^2+^ currents mediated by glycine receptors in OFF RGCs via the mediation of metabotropic P2Y(1) and P2Y(11) purinergic receptors, thus altering the pattern of action potentials in the retina [192]. Additionally, glycine release has been shown to be potentiated by cyclic AMP (cAMP) through exchange protein activated by cAMP2 (EPAC2) and Ca^2+^ stores in AII-type amacrine cells [193]. By using agonists and antagonists for EPAC2, Marc et al. found that the cAMP-induced increase in glycinergic release requires the activation of EPAC2 and intact internal Ca^2+^ stores [193]. cAMP level dynamics are an important part of glycine release within the retina, and it has been proposed that the activation of EPAC2 results in Ca^2+^ release from Ca^2+^ stores, which in turn increase the release of glycine onto OFF-cone BPCs terminals [193]. Such release modulation is important because it ensures proper synaptic transmission in the retina, allowing for sharper vision with reduced synaptic noise [253].

### 3.4. The Roles of Dopamine in Retinal Development, Normal Physiology, and Disease and Its Interplay with Calcium

Dopamine, being produced and released by dopaminergic amacrine cells in the retina, is largely responsible for retinal physiology and visual perception [122]. The neurotransmitter influences spatiotemporal vision, brightness, and the circadian clock, all of which contribute to the refinement and enhancement of our interpretation of the world [254].

#### 3.4.1. The Role of Dopamine in Retinal Development

Although retinal cells that will develop into dopaminergic neurons are detected early in retinal development, dopamine does not appear to contribute to the development of the retina or its synapses [122]. In mice, dopamine levels become detectable starting on postnatal day 6 and increase post-birth (Figure 9), when its function in relation to light-dark cycles is first observed [255,256].

#### 3.4.2. The Role of Dopamine in Normal Retinal Physiology

Dopamine plays a vital role in the normal functioning of the retina that is mediated primarily via the dopamine G protein-coupled receptors of types D_1_R and D_2_R, which influence cellular activity via the cAMP/PKA transduction pathway [123]. D_1_R enhances voltage-gated Ca^2+^ channels in neurons via stimulatory G proteins (G_s_), stimulating the release of dopamine from synaptic vesicles [122,123], while D_2_R utilizes inhibitory G proteins (G_i_) receptors to reduce dopamine release [123].

The synthesis and release of dopamine by amacrine cells are closely tied to the intensity of incident light on the retina, with dopamine levels increasing significantly during the day in response to light. In an opposing balance with melatonin-secreting cells, dopamine uses a feedback mechanism to modulate rod photoreceptor signaling, thereby aiding adaptation to changing light levels [122,257]. Interestingly, the response of dopamine to non-Circadian changes in the levels of light and darkness influences the receptive field properties of certain cells, enhancing their ability to detect specific visual features (Figure 10) [13]. With bright illumination (e.g., during daytime), dopamine release activates low-affinity D_1_ receptors on cone BPCs, strengthening their receptive field, which is hypothesized to enhance the detection of edges and fine details in the animal retina [124,125].

#### 3.4.3. The Role of Dopamine in Retinal Diseases

Visual impairments can be associated with diseases such as Parkinson’s disease (PD) that involve an imbalance in dopamine levels; in PD, there is a significant decrease in dopamine levels, including in the retina [137]. Individuals with PD can exhibit deficits in contrast sensitivity, loss of color vision, and reduced photopic and scotopic vision [136]. These symptoms are associated with the loss of dopaminergic amacrine and RGCs in the retina and impairment of electroretinographic response, suggesting that dopamine deficiency plays at least a partial role in the visual symptoms of PD [21,24].

Low levels of dopamine are also associated with form-deprivation myopia, a type of nearsightedness that occurs due to a developmental elongation of the eye [20,27], and in animal models, the administration of dopamine prevents the development of form-deprivation myopia [258,259]. Because light stimulates the release of dopamine, bright light exposure has been explored as a possible alternative treatment for form-deprivation myopia [260,261,262]. Thus, given the important interplay between light exposure and dopamine, it will be important to explore treatments and preventative methods to reduce visual impairment that combine dopamine and light.

#### 3.4.4. The Interplay between Dopamine and Calcium

While little is known about the roles of the interplay between Ca^2+^ and dopamine in development and retinal disease, dopamine receptors and Ca^2+^ work together in neurotransmission in the healthy retina. When activated, D4 receptors in photoreceptors induce changes in the elicited Ca^2+^ current that can increase or reduce neurotransmission in response to light, depending on the cell subtype [194,195]. The balance between melatonin and dopamine release also seems to be dependent on Ca^2+^. In the rabbit retina, Dubocovich (1983) reported that melatonin inhibits the Ca^2+^-dependent release of dopamine in a dose-dependent manner [196]. Similarly, Nowak et al. (1989) reported that melatonin blocks electrically evoked Ca^2+^-dependent dopamine release in the retina, suggesting that melatonin and dopamine, combined with Ca^2+^ dynamics, may regulate retinal mechanisms related to the Circadian rhythm [197].

### 3.5. The Roles of Acetylcholine in Retinal Development, Normal Physiology, and Disease and Its Interplay with Calcium

The activity of the neurotransmitter acetylcholine (ACh) in the CNS is critical for physiological activities such as direction selectivity, the regulation of retinal waves, and providing input to RGCs [202].

#### 3.5.1. The Role of Acetylcholine in Retinal Development

ACh plays a crucial role in retinal development, particularly in the context of stage II cholinergic retinal waves that are crucial for the establishment of the retinal circuitry and precede the glutamatergic waves in mice, occurring during postnatal days 1–10 [16,126], as shown in Figure 11. Just before the eyes of neonatal mice open, there is a shift from cholinergic connections to glutamatergic circuit activity, and retinal waves become regulated by glutamate, as described earlier in this review [19,144,263].

In RGCs, retinal waves occur in synchronized Ca^2+^ bursts that seem to coincide with the release of ACh [264,265]. The effects of ACh are closely linked to the activity of starburst amacrine cells (SACs), which use transient action potentials during development and may function as a part of a network that aids in the establishment of neural connectivity [127]. In various animals, the blockage of cholinergic waves is associated with changes in retinal development, such as a reduction in dendritic motility, stratification and growth in RGCs, and a reduction in the receptive field [266,267,268]. Arroyo et al. (2016) showed that the blockade of cholinergic retinal waves results in an increase in intrinsically photosensitive RGC gap junction networks with other neurons. This suggests that cholinergic waves normally suppress the formation of communication networks between these cells, supporting the interpretation that such network modulation is regulated through the dopaminergic release of these waves during retinal development [15]. Combined, such studies led to the hypothesis that ACh plays an important role in the development of the intricate retinal network, perhaps by contributing to the initiation and propagation of retinal waves, while also contributing to the growth and connectivity of retinal cells.

#### 3.5.2. The Role of Acetylcholine in Normal Retinal Physiology

ACh is produced from choline in starburst amacrine cells (SACs) and released. The effects of ACh in various cell types, including other amacrine cells, BPCs, and RGCs, are regulated by muscarinic acetylcholine receptors (mAChRs) and nicotinic acetylcholine receptors (nAChRs) [129,139,269,270]. Both mAChRs and nAChRs are present in BPCs, amacrine cells, and RGCs [129,200,271,272,273,274,275]. There are five subtypes of mAChRs: M1, M2, M3, M4, and M5 and five subtypes of nAChRs: α3, α4, α6, α7, β2, and β4 [200]. While mAChRs can enhance or diminish RGC responsiveness, nAChRs usually increase their responses by allowing the entry of Na^+^ and Ca^2+^ ions [129]. The activation of these receptors thus mediates the strength of the peak and inward current response of these cells to light stimuli (Figure 12), although the role of such changes is not clearly understood [129]. Additionally, the role of the retinal mAChRs in regulating functions such as pupil constriction, accommodation, and tear production has been the focal point of several studies [276]. Within the retina, the M_1_ subtype of mAchRs is especially important for the survival of RGCs [128].

Importantly, ACh plays a crucial role in direction selectivity, an important aspect of vision where direction-selective cells increase their activity in favor of specific orientations presented in the field of view [130]. The release of ACh from SACs occurs in a direction- and motion-sensitive manner and is transmitted to ON–OFF direction-selective RGCs [130,199,277,278]. These cells also secrete GABA, which is only released in response to null directions, thereby demonstrating an important interplay between GABA and ACh in determining direction selectivity [199]. Type 7 retinal BPCs normally contain nAChRs, which led to the investigation of the neuron’s role in the direction selectivity observed in SACs [198]. Type 7 BPCs isolated from mice with a genetic nAChR knockout were found to contribute to the ability of the SACs to respond to stimuli in a direction- and motion-dependent manner via cholinergic feedback [198]. This suggests that the cholinergic feedback to BPCs amplifies direction-selective signaling in SACs, highlighting the important role of ACh in augmenting motion detection in the retina [198].

#### 3.5.3. The Role of Acetylcholine in Retinal Diseases

ACh is not particularly associated with the development of retinal diseases; ACh receptors provide a promising avenue for neuroprotection from glaucoma and delaying the death of retinal cells. ACh can control neuroprotective pathways in both the developing and mature retina, particularly the mAchRs that are vital for neuronal survival [276]. This is supported by the observation that in cases of acute closed-angle glaucoma, the ACh agonist pilocarpine constricts the iris via the M_3_ receptors, thereby shielding the retina from long-term damage via vascular mechanisms [139]. However, it is important to mention that non-subtype-selective mAChR agonists can result in unwanted side effects. For example, pilocarpine has been used to control IOP, but its long-term use is associated with ciliary spasms, blurred vision, miosis, and retinal detachment [279], highlighting the need for the future development of treatments that maximize the benefits and minimize side effects.

#### 3.5.4. The Interplay between Acetylcholine and Calcium

As previously discussed, the Ca^2+^ bursts that occur during development propagate across the retina, where they coincide with the extracellular release of ACh, thereby contributing to the establishment of neuronal networks [198,199]. Thus, the interplay between Ca^2+^ and ACh is crucial in shaping retinal development. G protein-coupled mAChRs in the retina (including subtypes M1, M3, and M5) activate the alpha subunits of G_q/11_ proteins, raising the levels of intracellular Ca^2+^ [200]. This surge in Ca^2+^ can enhance PKC activity and potentially stimulate the production of nitric oxide (NO) via Ca^2+^-dependent neuronal nitric oxide synthase [200]. The M2 and M4 receptors are coupled to G_i_ and G_o_ proteins, respectively, and inhibit cAMP-dependent pathways, thereby indirectly influencing Ca^2+^ dynamics [200]. This intricate interplay between ACh and Ca^2+^ is a crucial regulatory mechanism in retinal function and signaling. The direction-selective release of ACh and GABA is a Ca^2+^-specific process and these neurotransmitters are likely released from different synaptic vesicle pools [199].

As discussed, although ACh is not associated with the development of many retinal diseases, some ACh agonists are neuroprotective. Drugs such as pilocarpine, which are effective against glaucoma, play neuroprotective roles by preserving the Ca^2+^ balance and the integrity of mitochondrial membranes [35].

## 4. Conclusions and Future Perspectives

In summary, the retina is crucial for the processing of visual information, with neurotransmitters and calcium dynamics being key factors in its function, development, and susceptibility to disease. This review highlighted the roles of the major retinal neurotransmitters, glutamate, GABA, glycine, dopamine, and ACh, shedding light on their functions and interactions and offering valuable insights for future research directions and the development of novel therapeutic agents to treat retinal diseases. Nevertheless, there remain major gaps in our understanding of the precise mechanisms and interactions in the retina, particularly concerning the interplay between neurotransmitters and Ca^2+^. For instance, antiepileptic drugs used to treat GABA-related retinal ischemia, such as ethosuximide, are often associated with reduced Ca^2+^ currents through T-type channels, and little is known about the Ca^2+^-dependent mechanisms of these drugs in relation to GABA. Also, although glycine, dopamine, and Ca^2+^ are important for different aspects of development and retinal disease, little research has addressed the dynamics of the interplay between each individual neurotransmitter and Ca^2+^. Another unclear understanding of the interplay between neurotransmitters and Ca^2+^ is observed with acetylcholine. nAChRs allow for the entry of Na^+^ and Ca^2+^, and the activation of nAChRs mediates the strength of the peak and inward current responses of retinal cells, but the purpose of these interactions remains poorly understood. Addressing such knowledge gaps will be crucial for future investigations. Exploring the early patterns and interactions between the neurotransmitters discussed in this review and Ca^2+^ during development and early life stages will contribute to an enhanced understanding of normal retinal functioning. Such knowledge will allow for the deeper exploration of early disease detection and advancement of various visual impairments.

## Figures and Tables

**Figure 1 ijms-25-02226-f001:**
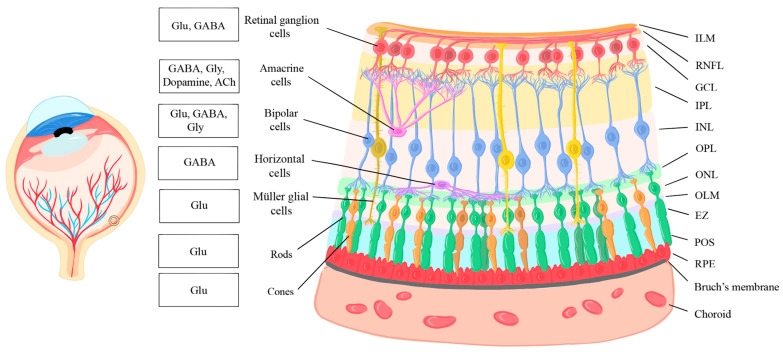
Schematic diagram of the eye, retina, retinal cell layers, and neurotransmitters found in the different cells. (**Left**): Schematic diagram of the eye where a section of the retina is highlighted by the circle, which is expanded in the right figure. (**Right**): Schematic diagram of retinal layers, retinal cells, and neurotransmitters. Abbreviations used: ACh, acetylcholine; EZ, ellipsoid zone; GCL, ganglion cell layer; Glu, glutamate; Gly, glycine; ILM, inner limiting membrane; INL, inner nuclear layer; IPL, inner plexiform layer; OPL, outer plexiform layer; ONL, outer nuclear layer; OLM, outer limiting membrane; POSs, photoreceptor outer segments; RNFL, retinal nerve fiber layer; RPE, retinal pigment epithelium.

**Figure 2 ijms-25-02226-f002:**
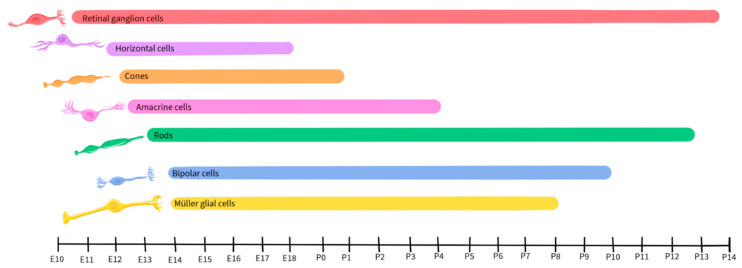
Timeline of retinal development in mice. Colored lines above the timescale show the time during embryonic and postnatal development when each retinal cell type arises. Abbreviations used: E, embryonic day; P, postnatal day.

**Figure 3 ijms-25-02226-f003:**
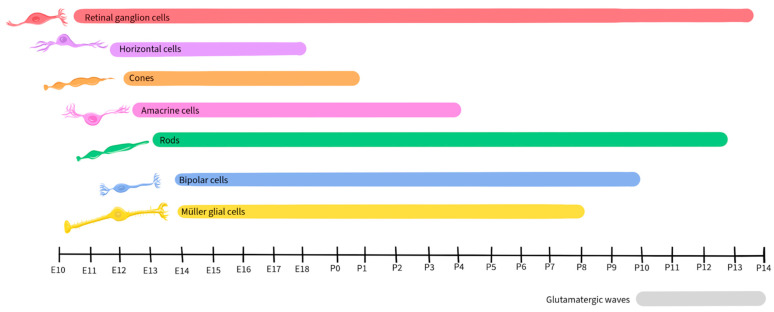
Timeline of retinal development and glutamate activity in mice. Colored lines above the timescale show the time during embryonic and postnatal development when each retinal cell type arises. Abbreviations used: E, embryonic day; P, postnatal day. The gray line below the timescale shows glutamatergic activity.

**Figure 4 ijms-25-02226-f004:**
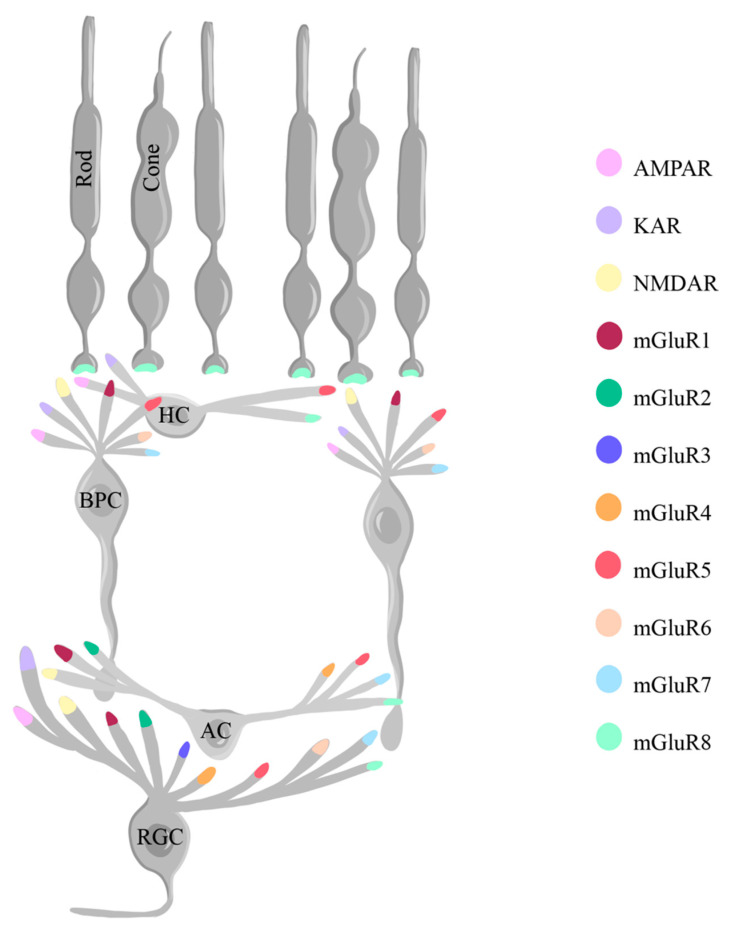
Localization of glutamate receptors in the retina. Left: grey illustrations depict the different retinal neurons and colored portions depict the expression of different glutamate receptors. Right: color legend indicating the types of glutamate receptors represented in the image on the left. Abbreviations: HC, horizontal cell; BPC, bipolar cell; AC, amacrine cell; RGC, retinal ganglion cell; mGluR, metabotropic glutamate receptor.

**Figure 5 ijms-25-02226-f005:**
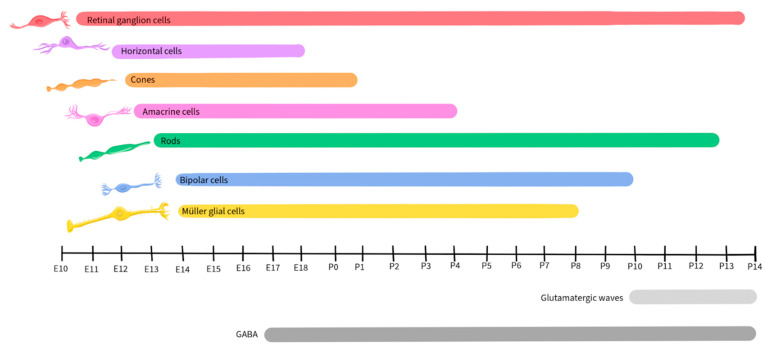
Timeline of retinal development and neurotransmitter activity in mice. Colored lines above the timescale show the time during embryonic and postnatal development when each retinal cell type arises. Abbreviations used: E, embryonic day; P, postnatal day. Gray lines below the timescale show glutamatergic and GABAergic activity.

**Figure 6 ijms-25-02226-f006:**
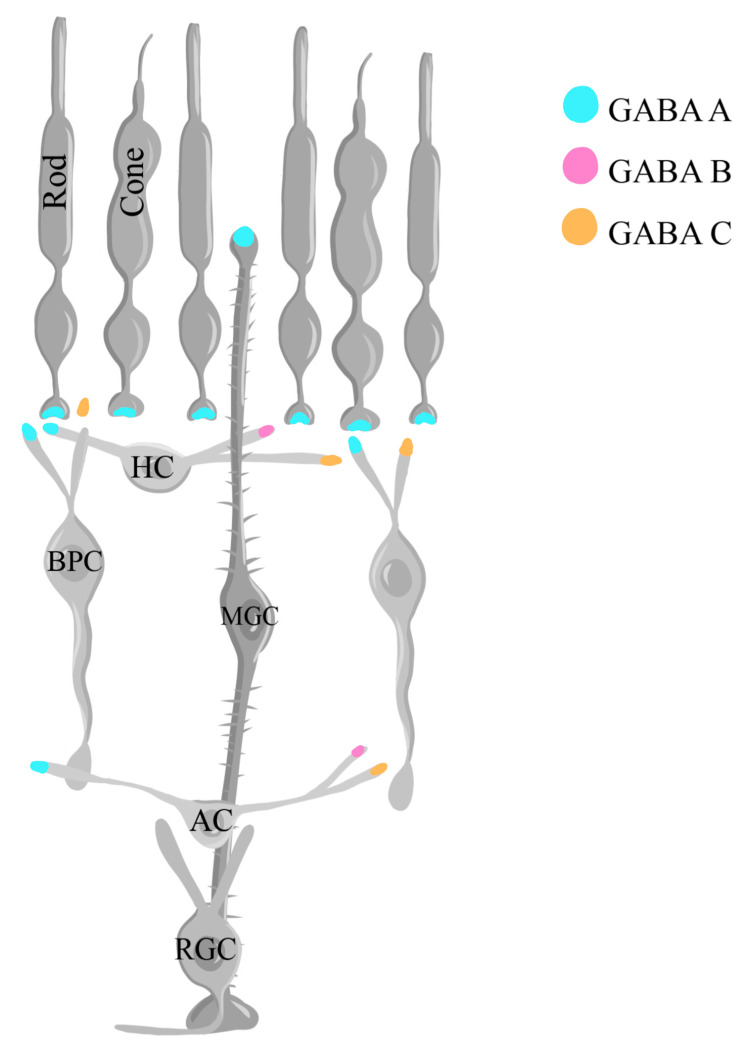
Localization of GABA receptors in the retina. Left: grey illustrations depict the different retinal neurons and colored portions depict the expression of different GABA receptors. Right: color legend indicating the types of GABA receptors represented in the image on the left. Abbreviations: HC, horizontal cell; BPC, bipolar cell; MGC, Müller glial cell AC, amacrine cell; RGC, retinal ganglion cell.

**Figure 7 ijms-25-02226-f007:**
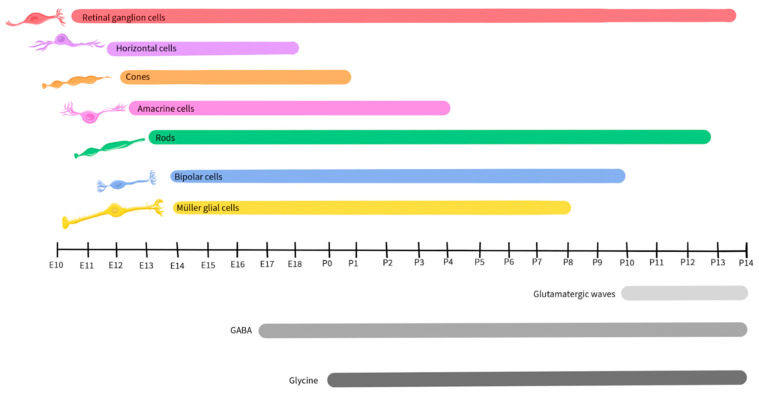
Timeline of retinal development and neurotransmitter activity in mice. Colored lines above the timescale show the time during embryonic and postnatal development when each retinal cell type arises. Abbreviations used: E, embryonic day; P, postnatal day. Gray lines below the timescale show glutamatergic, GABAergic, and glycinergic activity.

**Figure 8 ijms-25-02226-f008:**
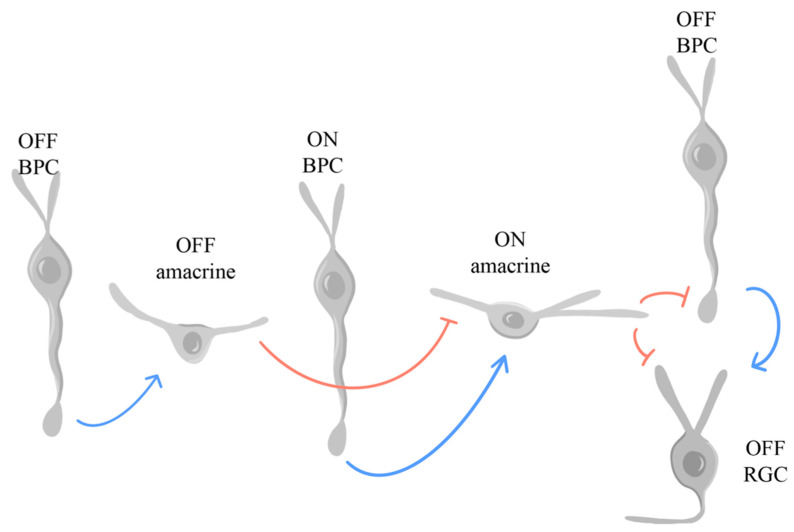
Diagram of glycine crossover inhibition in the retina adapted from Hsueh et al., 2008 [120]. Grey illustrations depict the different retinal neurons. Blue arrows represent glutamate excitation. Orange lines represent glycinergic inhibition. Abbreviations: BPC, bipolar cell; RGC, retinal ganglion cell.

**Figure 9 ijms-25-02226-f009:**
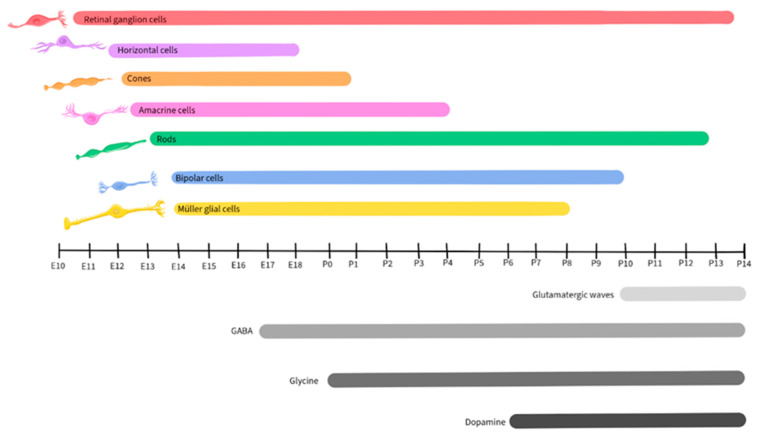
Timeline of retinal development and neurotransmitter activity in mice. Colored lines above the timescale show the time during embryonic and postnatal development when each retinal cell type arises. Abbreviations used: E, embryonic day; P, postnatal day. Gray lines below the timescale show glutamatergic, GABAergic, glycinergic, and dopaminergic activity.

**Figure 10 ijms-25-02226-f010:**
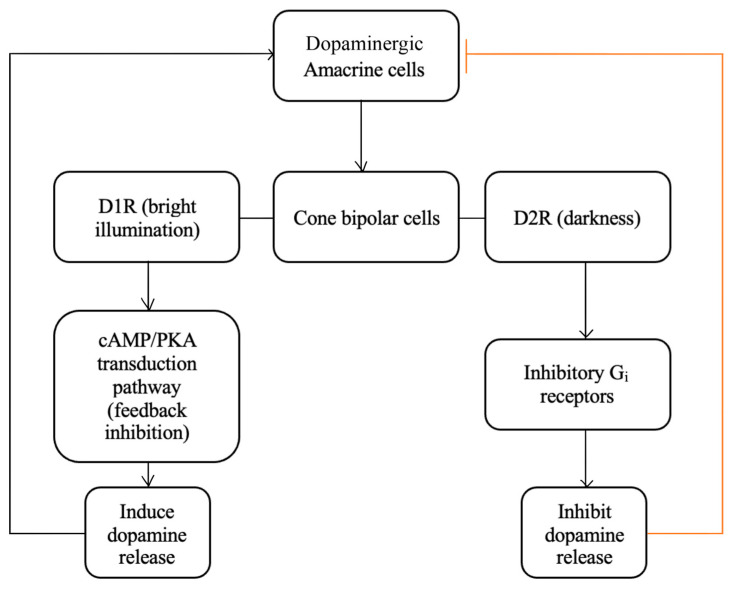
Flow diagram of dopamine transmission across the retina.

**Figure 11 ijms-25-02226-f011:**
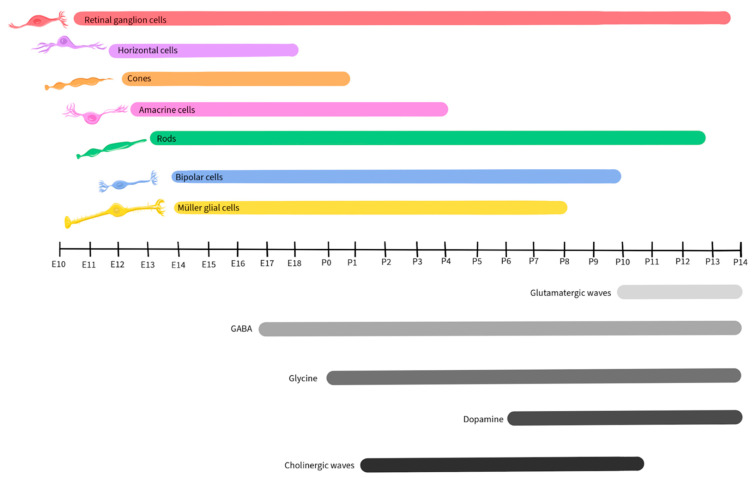
Timeline of retinal development and neurotransmitter activity in mice. Colored lines above the timescale show the time during embryonic and postnatal development when each retinal cell type arises. Abbreviations used: E, embryonic day; P, postnatal day. Gray lines below the timescale show glutamatergic, GABAergic, glycinergic, dopaminergic, and cholinergic activity.

**Figure 12 ijms-25-02226-f012:**
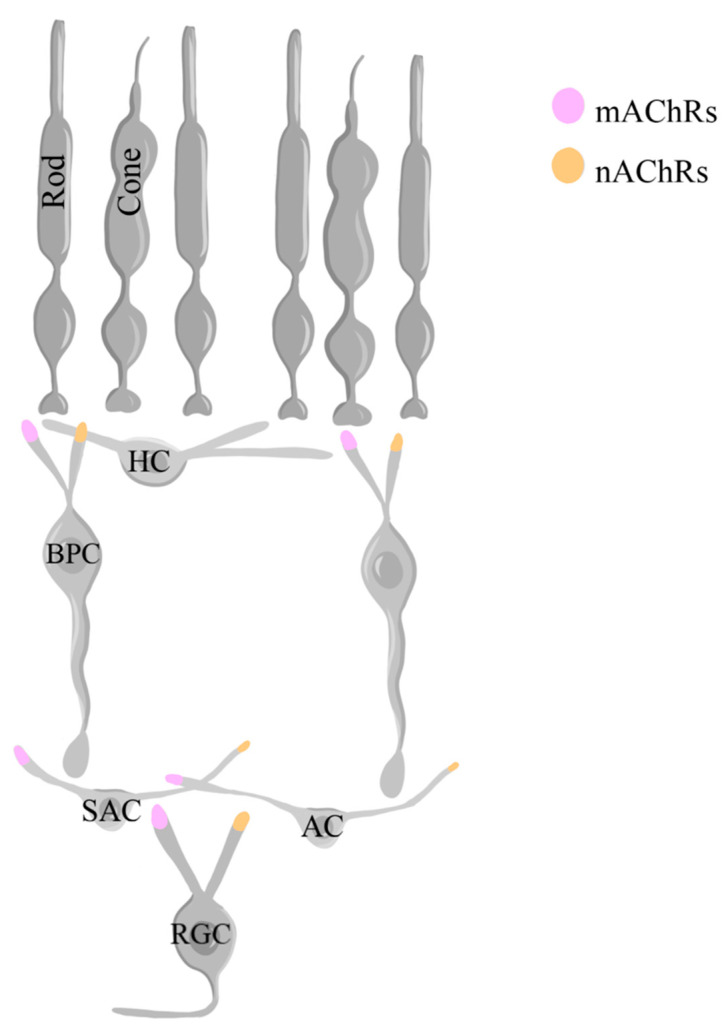
Localization of acetylcholine receptors in the retina. Left: grey illustrations depict the different retinal neurons and colored portions depict the expression of different ACh receptors. Right: color legend indicating the types of ACh receptors represented in the image on the left. Abbreviations: HC, horizontal cell; BPC, bipolar cell; SAC, starburst amacrine cell; AC, amacrine cell; RGC, retinal ganglion cell.

**Table 1 ijms-25-02226-t001:** Summary of neurotransmitter involvement in retinal development and healthy function.

Neurotransmitter	Retinal Development	Normal Retinal Physiology
Glutamate	Modulates retinal waves [18,101,102]. Stabilizes retinal ganglion cells (RGCs) dendrites regulated by NMDA receptors [102,103,104]. Regulates postembryonic development of RGCs [101,103].	Released upon photoreceptor depolarization, allowing for signal transmission and visual processing [105]. Regulation of ON/OFF bipolar cell (BPC) responses [106,107,108]. α-amino-3-hydroxy-5-methyl-4-isoxazolepropionic acid (AMPA) and kainite (KA) glutamate receptors mediate rapid synaptic transmission and allow for the passage of primarily sodium ions (Na^+^) and potassium ions (K^+^) in OFF BPCs, horizontal cells, and RGCs [109,110,111]. N-methyl-D-aspartate (NMDA) glutamate receptors mediate slower synaptic transmission and allow for the passage of Na^+^, K^+^, and calcium ions (Ca^2+^) in ON BPCs, amacrine cells, and RGCs [111].
GABA	Regulates calcium ion (Ca^2+^) waves in ganglion and amacrine cells prior to synapse formation [112]. Postnatal function of potentiating activity of RGCs during early development; later shifts to suppression of RGC burst activity [113].	GABA_A_ receptor inhibits proliferation and self-renewal of retinal progenitor cells [114]. GABA_B_ receptors play a role in modulating neurotransmitter release and the balance of brief and sustained signals in the retinal circuitry [115]. GABA_C_ receptors function as auto-receptors that regulate the release of GABA itself [116].
Glycine	Glycine receptors (GlyRs) contribute to the establishment of a depolarizing glycine-gated chloride ion (Cl^−^) flux, which triggers Ca^2+^ influx [117,118].	Serves as inhibitory neurotransmitter in most amacrine and some cone BPCs [119]. Inhibits ON and OFF pathways in inner nuclear layer (INL) by modulating the temporal responses and receptive field organization of RGCs [120]. In BPCs and A-type RGCs, α1β-containing GlyRs facilitate rapid signal transmission, with spontaneous inhibitory synaptic currents (sIPSCs) displaying medium fast kinetics with a lower decay time constant [119]. In AII amacrine cells, α3β-containing GlyRs relay rod light signals with reduced temporal resolution [119]. GlyR α1 subunits modulate visual signal transmission under scotopic conditions in the synapses of AII amacrine cells and OFF RGCs [121].
Dopamine	While present in early retinal development, it does not seem to be involved in the process [122].	Dopamine G-coupled receptor 1 (D_1_R) enhances voltage-gated Ca^2+^ channels via stimulatory G proteins, stimulating dopamine release [122,123]. Dopamine G-coupled receptor 2 (D_2_R) utilizes inhibitory G protein receptors to reduce dopamine release [123]. With bright illumination, dopamine release activates low-affinity D1 receptors on cone BPCs, strengthening their receptive field and therefore enhancing the detection of edges and fine details [124,125].
Acetylcholine (ACh)	Stage II cholinergic initiation and propagation of retinal waves prior to glutamatergic activity [16,126]. Establishment of neural activity linked to starburst amacrine cells (SACs) [127].	Important for the survival of retinal neurons, especially RGCs [128]. Activation of muscarinic acetylcholine receptors (mAChRs) and nicotinic acetylcholine receptors (NAChRs) mediates the strength of the peak and inward current response of retinal neurons to light stimuli [129]. Cholinergic feedback to BPCs amplifies direction-selective signaling in SACs [130].

**Table 2 ijms-25-02226-t002:** Summary of neurotransmitter involvement in retinal disease.

Neurotransmitter	Retinal Disease
Glutamate	Glaucoma [131,132,133]. Retinal ischemia [134,135]. Diabetic retinopathy [136].
GABA	Retinal ischemia [23]. Diabetic retinopathy [22].
Glycine	Diabetic retinopathy [28].
Dopamine	Parkinson’s disease visual impairment [137,138]. Form-deprivation myopia [20,21].
Acetylcholine (ACh)	Glaucoma [139].

## Data Availability

Not applicable.

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
