# Peer review of "The Interplay between Neurotransmitters and Calcium Dynamics in Retinal Synapses during Development, Health, and Disease"

_ijms, 2024, doi:10.3390/ijms25042226_

Round 1

Reviewer 1 Report

Comments and Suggestions for Authors

In this paper, interactions of the most well-characterized neurotransmitters (glutamate, GABA, glycine, dopamine, and acetylcholine) and changes in Ca2+ levels are reviewed in the retina. These interplays are discussed in ratinal development, physiology, and diseases as well. Retinal circuitry, synapses, and neurotransmisson are summarized in health and diseases in the first part of the paper followed by the specific interacton of each neurotransmitter and Ca2+homeostasis. Thus, these processes are discussed in the retina from several aspects based upon an exhaustive collection of literature references. The review is also based on well designed figures and informative tables.

Comments

Although these review stands as is, this Referee would suggest addition of some further information upon the Authors’ convenience.

2.2.2. GABAA receptors with different subunit (α1-5, β, γ) compositions turned to be a hot topic recently. It would worth to discuss this issue in the retina also as much as possible depending on data availability. This issue for glycine receptors is presented.

It would also be interesting to read about the presence and operation of the transporters of the various retinal neurotransmitters discussed. I particularly mention here GlyT1 and the contradicrory GlyT2 and the role of the former in ischemic and other retinal pathologies.

Minors

Line 146. Are the ON and OFF amacrine cells different in neurotransmitters they contain?

Line 238. GABA release

Table 2, line 2. Form-Deprivation

Page 9, line10. …the circadian rhythm

Fig. 10. Gi

Line 860. ACh

Line 865. antiepileptic drugs, specify.

Author Response

Reviewer 1

In this paper, interactions of the most well-characterized neurotransmitters (glutamate, GABA, glycine, dopamine, and acetylcholine) and changes in Ca2+ levels are reviewed in the retina. These interplays are discussed in retinal development, physiology, and diseases as well. Retinal circuitry, synapses, and neurotransmission are summarized in health and diseases in the first part of the paper followed by the specific interaction of each neurotransmitter and Ca2+homeostasis. Thus, these processes are discussed in the retina from several aspects based on an exhaustive collection of literature references. The review is also based on well-designed figures and informative tables.

Response: We appreciate the time and energy the reviewer committed and the value of their comments. We have incorporated our comments to the best of our knowledge.

Although this review stands as is, this Referee would suggest the addition of some further information at the Authors’ convenience.

  • 2.2. GABAAreceptors with different subunit (α1-5, β, γ) compositions turned out to be a hot topic recently. It would be worth discussing this issue in the retina also as much as possible depending on data availability. This issue for glycine receptors is presented.
    • Response: Thank you. We have now discussed the different subunits in section 2.2.2 as we judged relevant to the text. Please see lines 555-566.
  • It would also be interesting to read about the presence and operation of the transporters of the various retinal neurotransmitters discussed. I particularly mention here GlyT1 and the contradictory GlyT2 and the role of the former in ischemic and other retinal pathologies.
    • Response: Thank you. We have now added a paragraph discussing the glycine transporters. Unfortunately, little information is available concerning the transporters’ involvement in retinal pathologies. Please see lines 710-724.

Minors

  • Line 146. Are the ON and OFF amacrine cells different in neurotransmitters they contain?
    • Response: The literature does not mention any specific differences in the neurotransmitters that ON and OFF amacrine cells contain. The two different classifications are described to provide the reader with the different classifications based on light responses readers might encounter in literature. The main differences between neurotransmitters used by amacrine cells are not related to ON vs. OFF stratification but rather instead to the specific type of amacrine or, more generally, by their wide (most of them GABAergic) vs narrow (most of them Glycinergic) field morphology.
  • Line 238. GABA release
    • Response: Thank you. We have corrected it.
  • Table 2, line 2. Form-Deprivation
    • Response: Thank you. We have corrected it.
  • Page 9, line 10. …the circadian rhythm
    • Response: Thank you. We have corrected it.
  • 10. Gi
    • Response: Thank you. We have corrected it.
  • Line 860. ACh
    • Response: Thank you. We have corrected it.
  • Line 865. antiepileptic drugs, specify.
    • Response: Thank you. We have specified by adding the name of the drug we had previously mentioned in the body of the paper. Please see line 940

Reviewer 2 Report

Comments and Suggestions for Authors This is an excellent review that encompasses how the retina is formed and how it processes information. The representation of the timeline and retinal development in mice is outstanding, as is the observation of the activity of different neurotransmitters, along with the connection and modulation with calcium. The tables summarize the information described in the text adequately. However, I believe the authors should consider a few minor observations: 1. Figure 1 should have a title before the description of what is found on the right and left sides. 2. Table 2 needs better structuring. Perhaps just having two columns, one for the neurotransmitter and another for the diseases they are involved in, would suffice, as the role of certain neurotransmitters in many of these diseases is unknown. 3. Table 3 should be moved to after section 2.1 “Glutamate”. 4. I think that within the introduction section, the authors should include the definition of excitotoxicity. 5. Figures 5, 7, 9, and 11 are similar, only differing in how the activity of neurotransmitters is expressed. I recommend retaining only figure 11.

Author Response

Reviewer 2

This is an excellent review that encompasses how the retina is formed and how it processes information. The representation of the timeline and retinal development in mice is outstanding, as is the observation of the activity of different neurotransmitters, along with the connection and modulation with calcium. The tables summarize the information described in the text adequately. However, I believe the authors should consider a few minor observations:

Response: Thank you to this reviewer for their great interest. We appreciate the time and energy the reviewer committed and the value of their comments.

  • Figure 1 should have a title before the description of what is found on the right and left sides.
    • Response: Thank you. We have now added a title.
  • Table 2 needs better structuring. Perhaps just having two columns, one for the neurotransmitter and another for the diseases they are involved in, would suffice, as the role of certain neurotransmitters in many of these diseases is unknown.
    • Response: Thank you. We have updated Table 2 following the suggestions.
  • Table 3 should be moved to after section 2.1 “Glutamate”.
    • Response: Thank you. We have moved Table 3 to the suggested section.
  • I think that within the introduction section, the authors should include the definition of excitotoxicity.
    • Response: Thank you. We agree that we should define excitotoxicity for clarification. Instead of adding it to the introduction, we added the definition after mentioning the term for the first time with the goal of facilitating the understanding where relevant. Please see lines 430-431.
  • Figures 5, 7, 9, and 11 are similar, only differing in how the activity of neurotransmitters is expressed. I recommend retaining only Figure 11.
    • Response: Thank you. As the reviewer mentioned, we keep adding the relevant neurotransmitter to the subsection of the timeline of retinal development. We believe this will help the readers to follow the timelines instead of having it all in one figure. Thus, we choose to maintain all the images.

Reviewer 3 Report

Comments and Suggestions for Authors

This is an excellent review on the neurotransmitters and calcium dynamics in the retina. The review discusses the most important neurotransmitters in normal retinal development and physiological as well as pathological conditions. The tables and figures are also excellent. I only have a few minor comments. First of all, it is too long, some parts could be easily shortened, like the very general parts, for example sentences like the retina plays an important role in visual functions etc.... The OLM is missing from the description and only from Figure 1, only the ILM is there. Also in Figure 1 and 6, the Muller glial cells are depicted wrong, as the outer limiting membrane formed by the Muller glial cells is outside the photoreceptor nuclei, so the it should be one layer outside, not inner to the nuclei. I addition, in Figure 1, the arrow is not pointing at a Muller glial cell, but a photoreceptor cell. This should be corrected. The Table could contain the most important references.

Author Response

Reviewer 3

This is an excellent review on the neurotransmitters and calcium dynamics in the retina. The review discusses the most important neurotransmitters in normal retinal development and physiological as well as pathological conditions. The tables and figures are also excellent. I only have a few minor comments.

Response: Thank you to this reviewer for their great interest and constructive feedback on our manuscript. We have incorporated our comments to the best of our knowledge.

  1. First of all, it is too long, some parts could be easily shortened, like the very general parts, for example, sentences like the retina plays an important role in visual functions, etc...
    • Response: Thank you. We have reviewed the text in its entirety and shortened sentences as we judged appropriate.
  2. The OLM is missing from the description and only from Figure 1, only the ILM is there.
    • Response: Thank you. We have corrected figure 1.
  3. Also, in Figures 1 and 6, the Muller glial cells are depicted wrong, as the outer limiting membrane formed by the Muller glial cells is outside the photoreceptor nuclei, so it should be one layer outside, not the inner to the nuclei.
    • Response: Thank you. We have corrected Figures 1 and 6.
  4. In addition, in Figure 1, the arrow is not pointing at a Muller glial cell, but a photoreceptor cell. This should be corrected.
    • Response: Thank you. We have corrected figure 1.
  5. The Table could contain the most important references.
    • Response: Thank you. We have now added the most important references to all three tables.

Reviewer 4 Report

Comments and Suggestions for Authors

Reviewer report on Manuscript Draft entitled ‘The Interplay of Neurotransmitters and Calcium Dynamics in Retinal Synapses During Development, Health, and Disease’.

 In this Review manuscript, authors attempted overview important aspects of the retina, including its structure, synapses, visual processes, and development, and describe the roles of the retinal neurotransmitters glutamate, GABA, glycine, dopamine, and acetylcholine and their interplay with Ca2+ ions in retinal development, normal physiological functioning, and disease.

Presented manuscript and discussions are very valuable and very interesting from the point of view of molecular science. Manuscript is well designed and well supported by overview of literature. The research is in scope of the journal. Therefore, the manuscript can be published after some minor improvements and corrections:

 Introduction of the manuscript is not very informative, therefore it can be advanced by overview recent reviews on determination of Neurotransmitters and Neurodegenerative diseases (Molecularly Imprinted Polymers for the Recognition of Biomarkers for Some Neurodegenerative Diseases. Journal of Pharmaceutical and Biomedical Analysis, 2023, 228, 115343.).

 Titles of subchapters ‘2.1. Glutamate‘, ‘2.3. Glycine‘ and ‘2.4. Dopamine’ are not enough informative, therefore, these titles of subchapters could be formulated more efficiently.

Comments on the Quality of English Language

Minor editing of English language required.

Author Response

Reviewer 4

Reviewer report on Manuscript Draft entitled ‘The Interplay of Neurotransmitters and Calcium Dynamics in Retinal Synapses During Development, Health, and Disease’.

In this Review manuscript, authors attempted to overview important aspects of the retina, including its structure, synapses, visual processes, and development, and describe the roles of the retinal neurotransmitters glutamate, GABA, glycine, dopamine, and acetylcholine and their interplay with Ca2+ ions in retinal development, normal physiological functioning, and disease.

The presented manuscript and discussions are very valuable and very interesting from the point of view of molecular science. The manuscript is well-designed and well-supported by an overview of the literature. The research is in the scope of the journal. Therefore, the manuscript can be published after some minor improvements and corrections:

Response: Thank you to this reviewer for their great interest and constructive feedback on our manuscript.

  • The introduction of the manuscript is not very informative, therefore it can be advanced by an overview of recent reviews on the determination of Neurotransmitters and Neurodegenerative diseases (Molecularly Imprinted Polymers for the Recognition of Biomarkers for Some Neurodegenerative Diseases. Journal of Pharmaceutical and Biomedical Analysis,2023228, 115343.).
    • Response: Thank you. We have expanded some portions of the introduction to make it more informative.
  • Titles of subchapters ‘2.1. Glutamate‘, ‘2.3. Glycine‘ and ‘2.4. Dopamine’ are not enough informative, therefore, these titles of subchapters could be formulated more efficiently.
    • Response: Thank you. We have reformulated the titles of the subchapters and hope they are more informative.